# Groundwater Chemistry and Blood Pressure: A Cross-Sectional Study in Bangladesh

**DOI:** 10.3390/ijerph16132289

**Published:** 2019-06-28

**Authors:** Abu Mohd Naser, Thomas F. Clasen, Stephen P. Luby, Mahbubur Rahman, Leanne Unicomb, Kazi M. Ahmed, Solaiman Doza, Shadassa Ourshalimian, Howard H. Chang, Jennifer D. Stowell, K. M. Venkat Narayan, Mohammad Shamsudduha, Shivani A. Patel, Bethany O’Shea, Matthew O. Gribble

**Affiliations:** 1Department of Environmental Health, Emory University, Atlanta, GA 30322, USA; 2Emory Global Diabetes Research Center, Department of Global Health, Emory University, Atlanta, GA 30322, USA; 3Woods Institute for the Environment, Stanford University, Stanford, CA 94305, USA; 4International Centre for Diarrhoeal Disease Research, Bangladesh, Dhaka 1000, Bangladesh; 5Department of Geology, University of Dhaka, Dhaka 1000, Bangladesh; 6Department of Biostatistics and Bioinformatics, Emory University, Atlanta, GA 30322, USA; 7Institute for Risk and Disaster Reduction, University College London, London WC1E 6BT, UK; 8Department of Environmental and Ocean Sciences, University of San Diego, San Diego, CA 92110, USA

**Keywords:** medical geology, groundwater, chemical mixtures, blood pressure, groundwater, exposure mixtures, exposure combinations

## Abstract

*Background*: We assessed the association of groundwater chemicals with systolic blood pressure (SBP) and diastolic blood pressure (DBP). *Methods:* Blood pressure data for ≥35-year-olds were from the Bangladesh Demographic and Health Survey in 2011. Groundwater chemicals in 3534 well water samples from Bangladesh were measured by the British Geological Survey (BGS) in 1998–1999. Participants who reported groundwater as their primary source of drinking water were assigned chemical measures from the nearest BGS well. Survey-adjusted linear regression methods were used to assess the association of each groundwater chemical with the log-transformed blood pressure of the participants. Models were adjusted for age, sex, body mass index, smoking status, geographical region, household wealth, rural or urban residence, and educational attainment, and further adjusted for all other groundwater chemicals. *Results:* One standard deviation (SD) increase in groundwater magnesium was associated with a 0.992 (95% confidence interval (CI): 0.986, 0.998) geometric mean ratio (GMR) of SBP and a 0.991 (95% CI: 0.985, 0.996) GMR of DBP when adjusted for covariates except groundwater chemicals. When additionally adjusted for groundwater chemicals, one SD increase in groundwater magnesium was associated with a 0.984 (95% CI: 0.972, 0.997) GMR of SBP and a 0.990 (95% CI: 0.979, 1.000) GMR of DBP. However, associations were attenuated following Bonferroni-correction for multiple chemical comparisons in the full-adjusted model. Groundwater concentrations of calcium, potassium, silicon, sulfate, barium, zinc, manganese, and iron were not associated with SBP or DBP in the full-adjusted models. *Conclusions:* Groundwater magnesium had a weak association with lower SBP and DBP of the participants.

## 1. Introduction

Although macronutrients, micronutrients, and toxic chemicals co-occur in groundwater used for drinking purposes, few epidemiological studies have investigated the relationships of real-world drinking water chemical mixtures and blood pressure. The most common dissolved cations in groundwater are those abundant in the Earth’s crust, such as calcium, magnesium, sodium, and potassium [1]. These cations are also the essential macro-minerals for humans [2,3] that have an important role for the regulation of blood pressure [4,5].

There may be countervailing physiological impacts of different elements on blood pressure. In Bangladesh, 97% of the country’s rural population depends on groundwater for drinking purpose [6]. Bangladesh is the largest delta in the world formed by the deposition of sediments from the Himalayas by the Ganges, Brahmaputra, and Meghna rivers [7]. Sediments of different rivers have different chemical compositions: for example, Ganges-derived sediments have a higher calcium-magnesium carbonate content than the sediments of Brahmaputra and Meghna rivers [8]. Most of the groundwater in Bangladesh is of the CaHCO_3_ type, but Na-Ca-Mg-HCO_3_-Cl groundwater is abundant in salinity-affected coastal regions [9]. There has also been mass poisoning from high arsenic in groundwater in many areas of Bangladesh [10]. Drinking water arsenic concentration is associated with higher blood pressure in Bangladesh [11]. In coastal regions, high sodium intake through drinking groundwater has also been associated with high blood pressure in the adult population [12].

While chemicals co-occur in groundwater, few studies have adjusted other chemicals to evaluate the association of a specific chemical on population blood pressure. The objective of this analysis was to characterize the cross-sectional, potentially interacting, associations of local groundwater chemicals with blood pressure among groundwater-drinking adults aged ≥35-years-old in Bangladesh.

## 2. Materials and Methods

### 2.1. Data Sources and Participants

Data on individual-level demographic variables and health outcomes were derived from the Bangladesh Demographic and Health Survey (BDHS), which in 2011 recruited participants in 600 clusters representative of rural and urban areas of the 7 administrative divisions (regions) of Bangladesh. Each cluster was comprised of ~30 households [13]. The BDHS falls under the broader United States Agency for International Development (USAID) funded Demographic and Health Surveys (DHS), nationally-representative household surveys conducted in low- and middle-income countries approximately every 5 years to gather data on indicators of population health and nutrition [14]. In addition to the questionnaire data, BDHS surveyors collected data on anthropometry, test blood for anemia and HIV, and, in 2011 for the first time, blood pressure. One in three households of each cluster was randomly selected for blood pressure measurement [13]. All men and women ≥35-years-old in the selected households were eligible. Of the 7992 eligible adults contacted for blood pressure measurement, 105 refused to participate.

### 2.2. Blood Pressure Measures

Trained research staff used the LIFE SOURCE UA-767 Plus blood pressure automatic monitor to measure blood pressure for 7887 participants following the manufacturer’s recommended protocol. Blood pressure was measured among the ≥35-years-old male and female participants in one-third of the selected households in BDHS 2011 to determine the prevalence of hypertension among the adult population across the country. Appropriately-sized cuffs based on participants’ arm circumference were used for blood pressure measurement. Participants did not eat, drink caffeinated or carbonated drinks, or smoke within the 30 minutes prior to blood pressure measurement. There were 3 measurements taken at approximately 10-minute intervals, for both systolic blood pressure (SBP) and diastolic blood pressure (DBP), and the arithmetic mean of the second and third measurements were used in the analyses [13].

### 2.3. Clinical and Demographic Determinants of Blood Pressure

Age, sex, body mass index (BMI), educational attainment in years, participant smoking categories (current smoker versus not smoker), rural or urban residence, geographical region, and household wealth quintiles were compiled from the household questionnaire and biomarker assessment of the 2011 BDHS. The wealth index of the households was delivered in BDHS 2011 along with other variables. Additional information regarding the BDHS 2011 survey is provided in Appendix A [13].

### 2.4. Participant Drinking Water Source Data

Information on households’ primary drinking water sources were also obtained from BDHS 2011 dataset. We considered groundwater to be the primary source of drinking water if households reported using a tube well or borehole, or a protected or unprotected well, as their primary source of drinking water in the 2011 BDGS survey.

### 2.5. Hydro-Geological Data on Groundwater Chemistry

Well water chemical concentration data were obtained from the British Geological Survey (BGS), which collects groundwater information within the United Kingdom and internationally (Appendix A). BGS, in collaboration with the Department of Public Health Engineering (DPHE) of the government of Bangladesh, conducted a groundwater chemicals survey in 1998–1999 to develop maps showing the regional distribution of arsenic and other elements in the groundwater of Bangladesh, and to estimate the percentage of wells exceeding various limits for arsenic and other elements. Funded by the UK Department for International Development, the survey was conducted in two phases: the first phase (1998) covered the most arsenic-impacted southern and eastern districts and the second phase (1999) covered the remaining districts except the three districts in the Chittagong Hill Tracts [15]. Although these water chemistry data were collected several years before the DHS, the BGS-DPHE dataset is representative of the entirety of Bangladesh and includes rigorous laboratory measurement of different chemicals. The BGS-DPHE survey used stratified random sampling to ensure sampling sites were representative of the entire country, and collected 3534 well water samples across Bangladesh [16]. Samples were collected from wells ranging in depth from 7 to 362 meters, but 69% of the wells were in the depth interval 15-60 meter [17]. GPS coordinates of each well were collected in the BGS-DPHE survey. All samples were tested for arsenic, and all but 4 samples were tested for aluminum, boron, barium, calcium, cobalt, chromium, copper, iron, potassium, lithium, magnesium, manganese, sodium, silicon, sulfate, vanadium, and zinc in the BGS laboratories in UK. Arsenic was measured using hydride generation-atomic fluorescence spectrometry (HG-AFS); other chemicals were measured by inductively-coupled plasma-atomic emission spectrometry (ICP-AES) [16].

### 2.6. Selection of Chemicals for Analyses

The limit of detections (LODs) were 0.5 µg/L for arsenic; 0.01 mg/L for sodium and calcium; 0.008 mg/L for cobalt, chromium copper, lithium, and zinc; 0.006 mg/L for iron and vanadium; 0.5 mg/L for potassium; and 0.04 mg/L for magnesium [18]. The LODs were the values of chemicals in the BGS dataset that could be reliably measured by the analytical measurement procedures [19]. If any chemical had a value of <“x” in the dataset, “x” was considered as LOD (personal communication with BGS authority). Boron, cobalt, chromium, copper, and vanadium were excluded from analyses because more than half of those analyzed were below the limit of detection [18]. In addition, aluminum, lithium, and phosphorus were also excluded given many of their results were below the limit of detection. Although 28% of the samples had arsenic concentration below the limit of detection (<0.5 μg/L), we included arsenic in the analysis due to the importance of arsenic to the government of Bangladesh and the many previous epidemiological studies reporting an association of well water arsenic and blood pressure in Bangladesh [10,20]. Therefore, in our analysis, we included 11 groundwater chemicals—sodium, calcium, magnesium, silicon, potassium, barium, zinc, manganese, sulfate, iron, and arsenic. Additional information regarding the BGS-DPHE surveys are provided in Appendix A.

### 2.7. Drinking Water Chemical Exposure Assignment

Shapefiles of Bangladesh administrative units were obtained from DIVA-GIS (http://www.diva-gis.org/), a free online resource for shapefiles of different countries worldwide. We imported the GPS locations of the BDHS clusters and BGS-DPHE wells and projected them onto the Bangladesh shapefiles in ArcGIS 10.4.1 (ESRI, Redlands, CA, USA) using the UTM 1984 45 N projection system. In order to protect the identity of the households, one randomly selected GPS location was taken per BDHS 2011 cluster. We determined the nearest BGS-DPHE wells for each BDHS 2011 clusters using spatial joining in ArcGIS (Figure 1) and calculated the nearest distance in kilometers. We assigned the chemical exposure measures from the nearest BGS-DPHE well to each of the BDHS 2011 participants whose blood pressure was measured (i.e., the nearest well to their cluster, as BDHS geographic data were limited to cluster-level) as a surrogate of source water chemical concentrations contemporary to the blood pressure measures [18].

### 2.8. Statistical Analysis

In this study, we focused on participants who had blood pressure measurements and reported groundwater as the primary source of drinking water (*N* = 6875). Population means and standard deviations (SD) of the continuous variables, and population proportions of the categorical variables, were calculated using survey estimation methods. Concentrations of all water chemicals were right-skewed, and the 25th, 50th, and 75th percentiles of the water chemicals were reported. Spearman correlations were calculated between chemical pairs for all groundwater chemicals included in the analyses: sodium, calcium, silicon, magnesium, potassium, sulfate, iron, barium, zinc, manganese, and arsenic.

Survey estimation linear regression methods were used to estimate associations of water chemicals with log-transformed SBP or log-transformed DBP in the subpopulation who reported groundwater as the primary source of drinking water. The estimates were derived as geometric mean ratios (GMRs) due to use of log-transformed SBP and DBP. The estimates for one SD increase in each of the groundwater chemicals were derived. We sequentially fitted three models to assess the independent association of each water chemical with SBP and DBP. Model 1 estimated the association of each water chemical with blood pressure adjusted for age, sex, and BMI as continuous variables. Model 2 further adjusted for current smoking status (current smoker versus never of former smoker), education attainment (no institutional, ≤5 years, 6 to ≤10 years, and ≥11 years education), rural or urban residence, wealth score quintiles determined by BDHS, and regional location of the households. Model 3 further adjusted for all other chemicals in drinking water: sodium, calcium, silicon, magnesium, potassium, arsenic, sulfate, iron, manganese, barium, and zinc, using restricted cubic splines. We used Bonferroni correction to account for the multiple chemical adjustments in model 3 (α = 0.05/11). We predicted the differences of SBP and DBP for specific values of covariates in model 3 when concentrations for each chemical in groundwater increased from the 25th percentile to the 75th percentile and holding. For prediction, we included non-smoker females from Barisal region who were from the lowest wealth quintile and had no institutional education, and whose age, BMI, and all other chemical concentrations were set at mean values.

When a significant association was detected between a single chemical and a blood pressure outcome in any model, pairwise interactions were modeled to test whether there was significant effect modification by other chemicals after Bonferroni correction. Since the form of groundwater chemicals may vary across the surface geology [21], we also conducted stratified analyses by surface geological units of Bangladesh and conducted a meta-analysis of the estimates when significant associations were detected between a chemical and blood pressure.

Missing covariates (BMI for 247 participants and smoking status for 19 participants) were imputed by multiple imputation by chained equations [22]. We conducted a sensitivity analysis by restricting the BGS-DPHE wells to those located within 2.3 kilometers (mean value of distance between BDHS cluster and BGS-DPHE well) of the BDHS clusters. We reported results from all models in tables or figures, but results of model 3 are described in text in the results section. Statistical analyses were performed in Stata SE version 15.0 (StataCorp LLC, College Station, TX, USA) and graphs were prepared in R version 3.3.1 and ArcGIS version 10.4 (Esri, Redlands, CA, USA).

### 2.9. Ethical Approval

The study protocol was approved by Emory University IRB (IRB00088075) for the secondary data analyses. Informed consent was taken from the participants prior to blood pressure measurement in BDHS 2011. The DHS Program, which is the authority and compiles all DHS surveys for different countries and different years [23], provided the survey and GPS data. Permission was obtained from the copyright section of the British Geological Survey to use the publicly available BGS-DPHE dataset.

## 3. Results

### 3.1. Characteristics of the Participants

The mean age of the participants was 52 (95% CI: 51.4, 52.1), and the mean BMI was 20.6 kg/m^2^ (95% CI: 20.4, 20.7); 32% of participants were underweight, 57% were normal weight, 10% were overweight, and 2% were obese (Table 1). Nearly half of the population were male and 14% were active smokers. The mean years of schooling in this population was 3.1 years (95% CI: 2.9, 3.2). More than half of the population had no formal institutional education, 26% had primary, 17% had secondary, and 6% had college or higher-level education. Only 16.5% of the population resided in urban areas. The arithmetic mean SBP of the population was 118.7 mmHg (95% CI: 118.0, 119.4), and DBP was 77.9 mmHg (95% CI: 77.5, 78.5). The geometric mean of SBP was 116.9 mmHg (95% CI: 116.4, 117.4) and DBP was 78.8 mmHg (95% CI: 78.3, 79.4). SBP was higher among females, elderly, obese, and among college- or higher-level educated participants (Appendix A).

### 3.2. Distribution of Groundwater Chemicals

The median concentrations of groundwater chemicals from high to low order across all water samples were sodium (34.3 mg/L), calcium (25.4 mg/L), silicon (19.6 mg/L), magnesium (12.1 mg/L), potassium (3.0 mg/L), sulfate (0.8 mg/L), iron (0.7 mg/L), manganese (0.3 mg/L), barium (0.1 mg/L), zinc (0.01 mg/L), and arsenic (median: 3.3 µg/L). The WHO has not setup health-based guidelines for most of the chemicals we analyzed except barium (<700 µg/L) and arsenic (<10 µg/L) [24]. However, the median concentrations of all chemicals across Bangladesh were below the standard set by Bangladesh’s Department of Environment (Figure 2). Well water chemical concentrations varied by region. Sodium concentrations were higher in the three coastal regions (Barisal, Khulna, and Chittagong) when compared to non-coastal regions (Figure 2). The median calcium and magnesium concentration were high in Khulna region when compared to other regions. The magnesium concentration was below the Bangladesh drinking water standard in all regions (Figure 2). The median arsenic concentrations were below the Bangladesh standard in all regions. Potassium concentrations were relatively higher in coastal areas, including Chittagong and Barisal regions (Figure 2) than the other regions. Most of the groundwater chemical concentrations were positively correlated among each other. The Spearman correlation coefficients between sodium and magnesium was 0.61; sodium and potassium was 0.53; sodium and calcium was 0.13; and calcium and magnesium was 0.65 (Figure 3).

### 3.3. Association between Groundwater Chemicals and BP

In the full-adjusted model 3, one SD increase in groundwater sodium (162 mg/L) was associated with a 0.997 (95% CI: 0.990, 1.004) GMR of SBP and 0.991 (95% CI: 0.983, 0.999) GMR of DBP (Figure 4). One SD increase in groundwater calcium (44 mg/L) was associated with a 1.000 (95% CI: 0.988, 1.013) GMR of SBP and 0.995 (95% CI: 0.983, 1.006) GMR of DBP. One SD (19 mg/L) increase in well water magnesium (19 mg/L) was associated with a 0.984 (95% CI: 0.972, 0.997) GMR of SBP and 0.990 (95% CI: 0.979, 1.000) GMR of DBP. One SD (5 mg/L) increase in groundwater potassium was associated with a 1.007 (95% CI: 0.999, 1.014) GMR of SBP and 1.005 (95% CI: 0.998, 1.012) GMR of DBP. One SD (106 μg/L) increase in well water arsenic was associated with a 0.997 (95% CI: 0.991, 1.003) GMR of SBP and 0.999 (95% CI: 0.993, 0.994) GMR of DBP (Figure 4).

None of the groundwater chemicals, except magnesium, were associated with both SBP and DBP in all models (Figure 4). Groundwater sodium concentrations were associated with DBP in all models. Groundwater arsenic concentrations were associated with SBP and DBP in model 1, and with DBP in model 2. Groundwater concentrations of silicon, sulfate, barium, zinc, manganese, and iron were not associated with SBP and DBP in any models.

Increase in groundwater concentration from the 25th percentile to the 75th percentile distribution was associated with −0.15 (95% CI: −0.06, −0.25) SBP and −0.34 (95% CI: −0.25, −0.43) DBP for sodium; 0.06 (95% CI: 0.01, 0.11) SBP and −0.56 (95% CI: −0.66, −0.45) DBP for calcium; and −1.96 (95% CI: −2.25, −1.68) SBP and -0.84 (95% CI: −1.01, −0.67) DBP for magnesium in model 3. Increase in groundwater concentration from the 25th percentile to the 75th percentile distribution was associated with 0.47 (95% CI: 0.47, 0.48) SBP and 0.22 (95% CI: 0.22, 0.23) DBP for potassium; and −0.13 (95% CI: −0.12, −0.15) SBP and −0.03 (95% CI: −0.04, −0.02) DBP for arsenic (Table 2).

In sensitivity analyses conducted among wells located within 2.3 kilometers of BDHS clusters, we found similar weak effects of association between groundwater chemicals and blood pressure (Appendix A); however, confidence intervals were wider for most chemicals including magnesium on decreased sample size.

The pooled estimates from the meta-analysis of the surface geology suggest that a 10 mg/L increase in well water magnesium was associated with a 0.995 (95% CI: 0.989, 1.001) GMR of SBP and 0.994 (95% CI: 0.988, 1.000) GMR of DBP in Model 3 (Appendix A).

### 3.4. Interaction between Chemical Pairs to Influence BP

The effects of magnesium, calcium, sodium, and arsenic on SBP or DBP were not modified by the concentrations of any other chemicals in well water following Bonferroni correction (Table 3).

## 4. Discussion

We found that drinking water magnesium concentration in the well water was associated with lower SBP and DBP among adults aged 35 years and older in Bangladesh when adjusted for demographic variables, and additionally adjusted for socio-economic, educational attainments, and geographic locations of households. Groundwater sodium, potassium, and arsenic were associated with lower SBP and DBP when adjusted for demographic variables; however, their associations attenuated with further adjustments of covariates. Nevertheless, all identified associations were weak and associations of all chemicals, including magnesium, attenuated when adjusted for groundwater chemicals following Bonferroni correction.

The associations of drinking water magnesium with lower blood pressure is consistent with the findings from several studies. Epidemiological studies have highlighted the salubrious relationship between consumption of magnesium-rich foods and blood pressure [25,26]. A meta-analysis of 9 case-control studies found an overall negative association between drinking water magnesium and cardiovascular mortality [pooled odds ratio: 0.75 (95% Confidence Interval 0.68, 0.82)] [27]. In Israel, after adjustment for socio-demographic and clinical parameters, patients in desalinated areas, compared to non-desalinated areas, had lower blood magnesium concentrations (2.08 ± 0.27 vs. 1.94 ± 0.24 mg/dL, *p* < 0.001 from T-test) and higher all-cause mortality following hospitalization [hazard ratio: 1.87; 95% CI: 1.32–2.63] [28].

A typical diet in rural Bangladesh, containing seasonal local vegetables, small freshwater fish, and rice (balanced diet), is associated with optimum magnesium intake and lower systolic and diastolic blood pressure [29]. However, people consuming predominantly animal protein and root vegetables had lower intake of magnesium [29]. An analysis of 5256 wells in Bengal, Mekong, and Red River deltas, where 70% of the population relies on groundwater for drinking water, suggests that drinking water is an important source of daily intake of minerals. Individuals can obtain up to half of the daily recommended intake of magnesium from drinking two liters of groundwater in some areas of Bangladesh [30]. The magnesium-rich drinking water is associated with lower blood pressure in Bangladesh [31]. Magnesium in drinking water is highly bioavailable because it occurs in readily absorbable ionic forms [32].

Our analyses have several important limitations. Our drinking water chemical exposure data was one decade older than the blood pressure outcome data. There may be substantial exposure measurement errors resulting from an inability to adjust for the temporal variability of water chemistry data. Chemical concentrations in groundwater have temporal patterns, and chemicals may not all shift in the same direction or by the same magnitude over time. In China and Bangladesh, greater annual temporal variation was observed for redox-sensitive chemicals, such as Fe, Mn, and As [33,34]. Temporal variation may also depend on the depth of the water: in Bangladesh, the major chemicals in groundwater (Na, K, Mg, Ca, and Cl) from shallow wells (<30 m deep) varied around ± 90% of baseline concentration over a period of 2–3 years, however, these chemicals varied <10% in deeper wells (>30 m) [33]. The instability of chemical concentrations over time in shallow wells may be due to greater interaction with freshwater (e.g., rainwater) [33]. The median depth of the BGS-DPHE wells were 35 meters (IQR: 22, 56), so participants consuming drinking water from the shallow groundwater will have more temporal variability of groundwater chemicals and differential exposure measurement errors than participants who consume deep groundwater.

Although studies in Bangladesh have identified that drinking water sodium and arsenic are associated with high blood pressure [11], we found that sodium and arsenic had no association with high blood pressure or association with low blood pressure when adjusted for other chemicals. Exposure measurement errors may also explain why our findings are inconsistent with other epidemiological studies conducted in Bangladesh. Since the BGS survey conducted in 1999, many deep tube wells (>150 m) have been installed in Bangladesh [35], and participants likely consumed groundwater from deeper wells with lower arsenic concentrations when their blood pressure was measured during the 2011 BDHS survey. Groundwater sodium concentrations are high during the dry season in seawater intrusion-affected coastal Bangladesh. Studies highlighting the association between drinking water sodium and blood pressure were conducted in coastal Bangladesh during the dry season. Nevertheless, the BDHS survey was conducted during the monsoon and post-monsoon season when groundwater sodium was not high. We lack the time of water sample collection during the BGS survey, which limits our understanding of well water concentrations due to seasonal variability.

A second source of exposure measurement error may be the misclassification of the nearest wells to individual participants. In Bangladesh, small-scale spatial variation of groundwater chemicals exists [36,37,38]. Average distance to the nearest BGS-DPHE well was 2.3 kilometers for our study participants, and it is possible that actual household wells had different chemical concentrations than the nearest selected BGS-DPHE wells. We believe this could lead to a non-differential exposure misclassification, which can magnify the bias for the continuous blood pressure outcome [39]. Another source of information bias is the cross-sectional single-time measurement of blood pressure, which has diurnal variation. Participants whose blood pressure were measured during the afternoon can have higher blood pressure than those whose blood pressure was measured in the morning; however, the BDHS data do not have information regarding time of blood pressure measurement. Other factors such as nutrient and mineral intake through diet, physical activities, and sleep patterns can influence blood pressure, but we were unable to adjust for those factors. Nevertheless, participants’ socio-economic status and rural/urban residence may be proxies for some factors, such as diet and physical activities to some extent.

## 5. Conclusions

We found weak associations between water chemicals and blood pressure; however, exposure measurement errors may have biased our findings. Precise measurements of groundwater chemical mixtures will better detect the associations of any chemicals on blood pressure and interactions if present.

## Figures and Tables

**Figure 1 ijerph-16-02289-f001:**
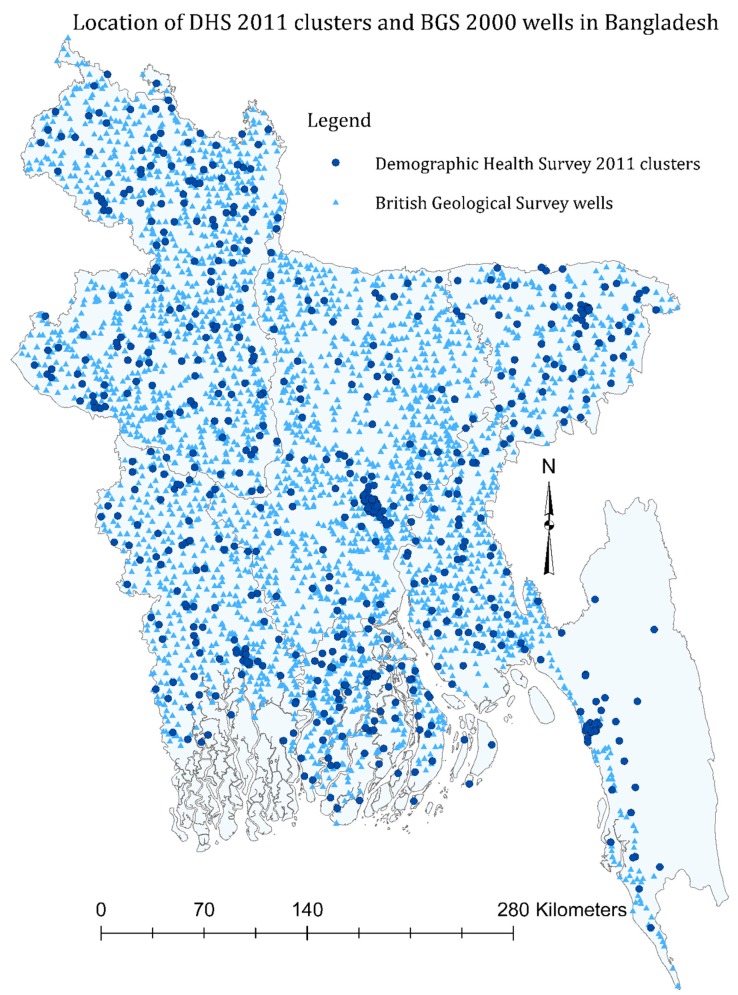
Location of demographic and health survey (DHS) 2001 clusters and British Geological Survey (BGS) 1998–1999 wells in Bangladesh.

**Figure 2 ijerph-16-02289-f002:**
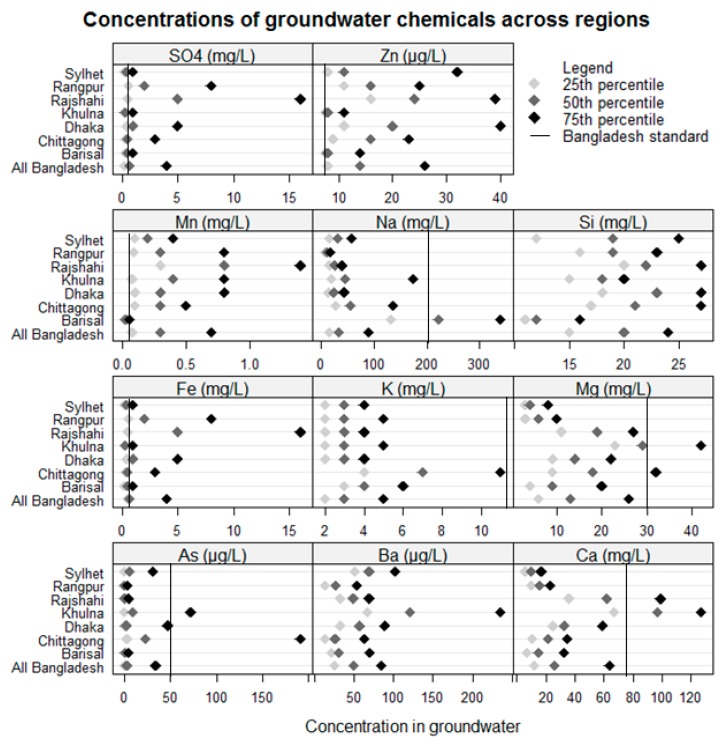
Concentration of groundwater chemicals across regions.

**Figure 3 ijerph-16-02289-f003:**
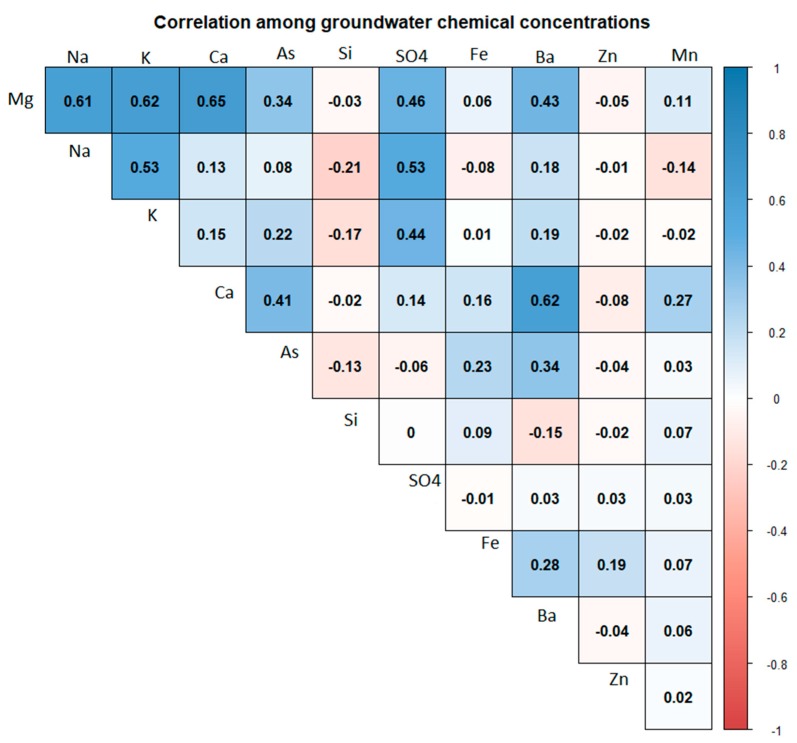
Spearman correlations between groundwater chemical pairs in BGS- Department of Public Health Engineering (DPHE) wells.

**Figure 4 ijerph-16-02289-f004:**
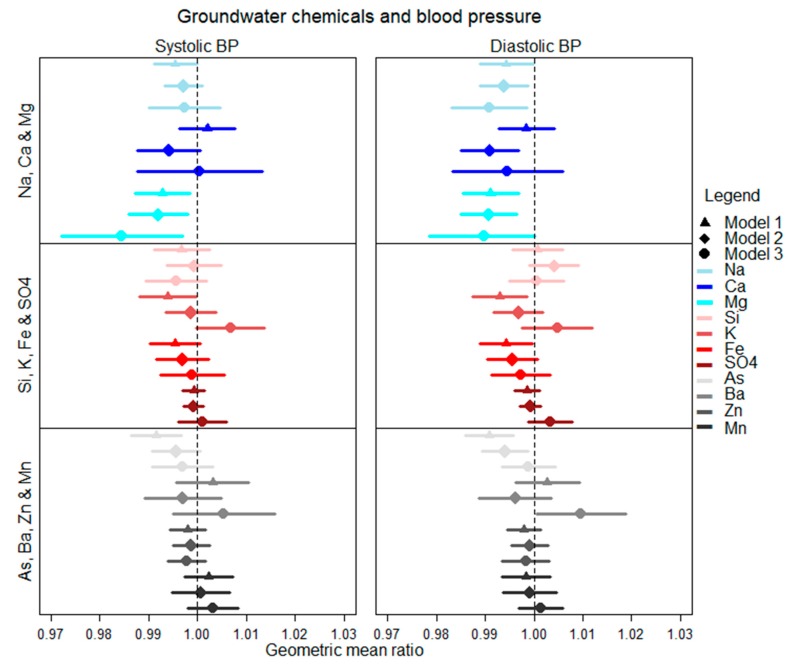
Associations between groundwater chemicals and blood pressure. Associations are shown for one standard deviation increase in water chemicals.

**Table 1 ijerph-16-02289-t001:** Characteristics of the ≥35-year-old participants with blood pressure measurements in the Bangladesh Demographic and Health Survey (BDHS) 2011.

Participants’ Characteristics	All Participants with Blood Pressure Measurement (*N* = 7887)	Groundwater Drinkers with Blood Pressure Measurement (*N* = 6875)	Participants Eligible for Blood Pressure Measurement (*N* = 7992)
Age, mean (95% Confidence Interval)	51.4 (51.1, 51.7)	51.7 (51.4, 52.1)	51.5 (51.1, 51.8)
BMI (kg/m^2^), mean (95% CI)	20.9 (20.7, 21.0)	20.6 (20.4, 20.7)	20.9 (20.7, 21.0)
BMI categories, %			
Underweight (<18.5 kg/m^2^)	29.1 (27.7, 30.6)	31.1 (29.6, 32.6)	29.1 (27.7, 30.6)
Normal weight (≥18.5 to <25 kg/m^2^)	57.4 (56.0, 58.8)	57.3 (55.9, 58.9)	57.4 (56.0, 58.8)
Overweight (≥25 to <30 kg/m^2^)	11.3 (10.5, 12.3)	9.9 (9.0, 10.9)	11.4 (10.5, 12.3)
Obese (≥30 kg/m^2^)	2.1 (1.8, 2.5)	1.7 (13.7, 2.1)	2.1 (1.8, 2.5)
Years of education, mean (95% CI)	3.3 (3.3, 3.5)	3.1 (2.9, 3.2)	3.4 (3.2, 3.6)
Education categories, % (95% CI)			
No institutional education	49.6 (47.7, 51.4)	52.0 (50.0, 53.9)	49.6 (47.7, 51.4)
Primary level (≤5 years)	25.8 (24.4, 27.1)	25.9 (24.5, 27.3)	25.8 (24.4, 27.1)
Secondary level (6 to ≤10 years)	17.5 (16.3, 18.7)	16.7 (15.5, 17.9)	17.5 (16.3, 18.7)
College level or higher (≥11 years)	6.9 (6.0, 7.9)	5.5 (4.7, 6.3)	7.2 (6.2, 8.3)
Male sex, % (95% CI)	49.4 (48.6, 50.2)	49.4 (48.5, 50.3)	49.5 (48.7, 50.2)
Female sex, % (95% CI)	50.6 (49.8, 51.3)	50.6 (49.7, 51.5)	50.5 (49.7, 51.3)
Current Smoker, % (95% CI)	13.6 (12.2, 15.1)	13.8 (12.3, 15.3)	13.6 (12.2, 15.1)
**Household characteristics**			
Urban residence, % (95% CI)	23.3 (22.2, 24.4)	16.5 (15.0, 18.1)	23.8 (22.8, 25.0)
Wealth index, (95% CI)			
Quintile 1	19.4 (17.7, 21.3)	20.8 (18.9, 22.8)	19.3 (17.5, 21.1)
Quintile 2	19.2 (17.8, 20.7)	21.1 (19.6, 22.8)	19.1 (17.7, 20.6)
Quintile 3	19.8 (18.3, 12.4)	21.5 (19.9, 23.2)	19.8 (18.2, 21.3)
Quintile 4	20.7 (19.1, 22.3)	21.4 (19.8, 23.2)	20.6 (19.0, 22.2)
Quintile 5	20.9 (19.2, 22.7)	15.1 (13.5, 16.9)	21.3 (19.6, 23.2)
**Cluster characteristics**	% (*n*/*N*)		
Distance of nearest well in kilometers, mean (95% CI)	3.2 (2.8, 3.7)	3.3 (2.9, 3.8)	3.2 (2.8, 3.7)
Divisional distribution, % (95% CI)			
Dhaka	32.1 (30.9, 33.3)	29.0 (27.3, 30.7)	32.3 (31.1, 33.6)
Barisal	5.9 (5.5, 6.4)	6.3 (5.8, 6.9)	5.9 (5.5, 6.4)
Chittagong	17.0 (16.1, 17.9)	17.6 (16.6, 18.6)	16.9 (16.0, 17.8)
Khulna	13.0 (12.3, 13.8)	12.9 (11.9, 14.0)	12.9 (12.2, 13.7)
Rajshahi	14.5 (13.6, 15.4)	15.5 (14.5, 16.6)	14.6 (13.7, 15.5)
Rangpur	11.7 (11.1, 12.4)	12.9 (12.2, 13.7)	11.7 (11.1, 12.3)
Sylhet	5.7 (5.3, 6.1)	5.8 (5.3, 6.3)	5.8 (5.4, 6.2)

**Table 2 ijerph-16-02289-t002:** Prediction of change in systolic blood pressure (SBP) and diastolic blood pressure (DBP) when chemicals increased from the 25th percentile to the 75th percentile distribution. Predictions were done for non-smoker females from Barisal region who were from the lowest wealth quintile and had no institutional education, and whose age, BMI and all other chemical concentrations were set at mean values.

Chemicals	25th Percentile	75th Percentile	Predicted Change in BP	
			SBP (95% CI)	DBP (95% CI)
Na (mg/L)	15	91	−0.15 (−0.06, −0.25)	−0.34 (−0.25, −0.43)
Ca (mg/L)	13	71	0.06 (0.01, 0.11)	−0.56 (−0.66, −0.45)
Mg (mg/L)	7	27	−1.96 (−1.68, −2.25)	−0.84 (−1.01, −0.67)
K (mg/L)	2	5	0.47 (0.47, 0.48)	0.22 (0.22, 0.23)
Fe (mg/L)	0.12	4.28	−0.14 (−0.06, −0.21)	−0.22 (−0.12, −0.32)
Si (mg/L)	15	24	−0.77 (−0.97, −0.77)	0.06 (−0.09, 0.22)
S04 (mg/L)	0.2	3.7	0.01 (0.00, 0.02)	0.02 (0.02, 0.02)
As (μg/L)	0.5	39	−0.13 (−0.12, −0.15)	−0.03 (−0.04, −0.02)
Ba (μg/L)	26	88	0.38 (0.37, 0.39)	0.45 (0.45, 0.45)
Zn (μg/L)	8	25	−0.03 (−0.04, −0.03)	−0.02 (−0.02, −0.010
Mn (μg/L)	76	735	0.29 (0.26, 0.31)	0.09 (0.07, 0.10)

**Table 3 ijerph-16-02289-t003:** Tests of pair-wise interactions between groundwater chemicals in relation to blood pressure outcomes. Associations were Bonferroni-significant at α = 0.005.

Chemical Pairs	*p* Values		Chemical Pairs	*p* Values	
	Systolic BP	Diastolic BP		Systolic BP	Diastolic BP
Mg & Ca	0.318	0.185	Na & Si	0.307	0.900
Mg & Na	0.854	0.307	Na & Mn	0.217	0.827
Mg & K	0.811	0.386	Na & Zn	0.772	0.566
Mg & As	0.593	0.421	Na & Fe	0.089	0.045
Mg & Si	0.308	0.796	Na & SO_4_	0.012	0.591
Mg & Ba	0.668	0.102	Na & Ba	0.393	0.024
Mg & Zn	0.684	0.583	Na & Ca	0.503	0.254
Mg & Fe	0.820	0.109	Na & Mg	0.938	0.958
Mg & SO_4_	0.231	0.034	Na & K	0.814	0.811
Mg & Mn	0.755	0.782	Na & As	0.188	0.014
Ca & Si	0.250	0.750	K & Si	0.440	0.529
Ca & Mn	0.537	0.368	K & Mn	0.254	0.340
Ca & Zn	0.198	0.225	K & Zn	0.899	0.423
Ca & Fe	0.678	0.058	K & Fe	0.499	0.304
Ca & SO_4_	0.279	0.073	K & SO_4_	0.849	0.277
Ca & Ba	0.121	0.003	K & Ba	0.750	0.083
Ca & Mg	0.013	0.026	K & Ca	0.258	0.007
Ca & Na	0.604	0.670	K & Mg	0.452	0.582
Ca & K	0.641	0.417	Na & K	0.295	0.987
Ca & As	0.819	0.917	K & As	0.819	0.953
As & Si	0.012	0.184	As & Ba	0.991	0.430
As & Mn	0.672	0.285	As & Ca	0.710	0.294
As & Zn	0.568	0.890	As & Mg	0.361	0.801
As & Fe	0.387	0.911	As & K	0.564	0.266
As & SO_4_	0.763	0.348	AS & Na	0.113	0.199

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
