# Peer review of "Groundwater Chemistry and Blood Pressure: A Cross-Sectional Study in Bangladesh"

_ijerph, 2019, doi:10.3390/ijerph16132289_

Round 1

Reviewer 1 Report

This MS explores the association of groundwater chemicals in Bangladesh with SBP and DBP in individuals. It is well organized and easy to follow. Although the ethical approval was stated in the MS did participants gave informed consent for the study? Participants were of both gender? Please clarify (Table 1).

Line 93 – Since formula of BMI is well known please remove “weight and height as kg/m2” in the sentence “BMI was calculated.

Line 129 – Please rephrase: “In addition, aluminum, 129 lithium and … went undetected.

Author Response

Thank you very much for reviewing our manuscript titled “Groundwater chemistry and blood pressure: a cross-sectional study in Bangladesh” and providing us the opportunity to revise the manuscript based on your comments. Please find our point-by-point responses below (in red text).

Point 1: This MS explores the association of groundwater chemicals in Bangladesh with SBP and DBP in individuals. It is well organized and easy to follow. Although the ethical approval was stated in the MS did participants gave informed consent for the study? Participants were of both gender? Please clarify (Table 1).

Response 1: Thanks for your comment. Emory University IRB approved the secondary data analyses. Informed consent was taken from the participants prior to blood pressure measurement in BDHS 2011. Participants were of both gender. We have added a line in Table 1 to clarify this.

Point 2: Line 93 – Since formula of BMI is well known please remove “weight and height as kg/m2” in the sentence “BMI was calculated.

Response 2: We have deleted that line as per the suggestion.

Point 3: Line 129 – Please rephrase: “In addition, aluminum, 129 lithium and … went undetected.

Response 3: We have rephrased the sentence as suggested.

Reviewer 2 Report

Manuscript ID: International Journal of Environmental Reseach and Public Health;

ijerph-526925

This manuscript is very well written, thorough, and informative. Below are two minor suggestions for the authors' considerations:

Line 64: Describe reasoning or justification for focusing on adults 35 years of age and older. For example, is that based on relevance to blood pressure prevelance data?

Line 84: What about physical activity? Were the participants also instructed not to engage in physical activities/exercise that could elevate their typical blood pressure?

Line 272: Discussion section. Other study limitations, such as potential influence of diet and exercise habits on blood pressure of survey participants could be addressed in the discussion. For example, could the influence of these factors be captured indirectly through the socioeconomic status of participating households?

Author Response

Thank you very much for reviewing our manuscript titled “Groundwater chemistry and blood pressure: a cross-sectional study in Bangladesh” and providing us the opportunity to revise the manuscript based on your comments. Please find our point-by-point responses below (in red text).

Point 1: This manuscript is very well written, thorough, and informative. Below are two minor suggestions for the authors' considerations:

 Line 64: Describe reasoning or justification for focusing on adults 35 years of age and older. For example, is that based on relevance to blood pressure prevelance data?

 Response 1: The blood pressure measurement of adults 35 years of age and older were implemented to determine the prevalence of hypertension among the adults in Bangladesh. We have clarified this in the method section.  

Point 2: Line 84: What about physical activity? Were the participants also instructed not to engage in physical activities/exercise that could elevate their typical blood pressure?

Response 2: Thank you very much for the comment. We agree that physical activities could potentially elevate participants blood pressure. Nevertheless, the BDHS 2011 report did not mention about screening of physical activity within 30 minutes of blood pressure measurement. We have clarified this limitation in the discussion section of the revised manuscript.

 Pont 3: Line 272: Discussion section. Other study limitations, such as potential influence of diet and exercise habits on blood pressure of survey participants could be addressed in the discussion. For example, could the influence of these factors be captured indirectly through the socioeconomic status of participating households?

Response 3: Thank you for your comment. We have highlighted additional limitations in the discussion section of the revised manuscript.